# Effect of Titanium Cation Doping on the Performance of In₂O₃ Thin Film Transistors Grown via Atomic Layer Deposition

Bing Yang [1,2], Pingping Li [1,3], Zihui Chen [1,2], Haiyang Xu [1], Chaoying Fu [4,*], Xingwei Ding [1,3,*] and Jianhua Zhang [1,2]

1 Key Laboratory of Advanced Display and System Application, Ministry of Education, Shanghai University, Shanghai 200072, China
2 School of Microelectronics, Shanghai University, Shanghai 200072, China
3 School of Mechatronics and Automation, Shanghai University, Shanghai 200072, China
4 Huzhou Key Laboratory of Medical and Environmental Applications Technologies, School of Life Sciences, Huzhou University, Huzhou 313000, China
* Correspondence: 03064@zjhu.edu.cn (C.F.); xwding@shu.edu.cn (X.D.)

**Abstract:** Indium oxide semiconductors, as one of the channel materials for thin film transistors (TFTs), have been extensively studied. However, the high carrier concentration and excess oxygen defects of intrinsic In₂O₃ can cause the devices to fail to work properly. We overcame this hurdle by incorporating the titanium cation ($Ti^{4+}$) into In₂O₃ via atomic layer deposition (ALD). The InTiO$_x$ TFTs with an In:Ti atomic ratio of 15:1 demonstrated excellent electrical and optical properties, such as a lower threshold voltage ($V_{th}$) of 0.17 V, a lower subthreshold swing ($SS$) of 0.13 V/dec., a higher $I_{on}/I_{off}$ ratio of $10^7$, and a transmittance greater than 90% in the visible region. With the doping ratio increasing from 20:1 to 10:1, the mobility decreased from 9.38 to 1.26 cm²/Vs. The threshold voltage shift ($\Delta V_{th}$) of InTiO (15:1) under 5 V positive bias stress (PBS) for 900 s is 0.93 V, which is less than other devices. The improvement in stability with increasing $Ti^{4+}$ concentrations is attributed to the reduction of oxygen defects. Therefore, these InTiO (15:1) TFTs with excellent performance show great potential for future applications in transparent electronic devices.

**Keywords:** atomic layer deposition; Ti-doped In₂O₃; oxygen defects; thin film transistors





## 1. Introduction

Recently, amorphous oxide semiconductors (AOS) have received extensive attention in the field of flat panel displays and wearable devices due to their excellent photoelectric properties [1–3]. Among the various candidate materials, indium oxide (In₂O₃) with a wide bandgap (3.6–3.7 eV) has a high transmittance to visible light and exhibits excellent chemical stability [4–6], making it one of the most promising thin film transistor channel materials. In-ions with an outer electronic structure of $(n-1)d^{10}ns^0$ ($n \geq 4$) [7] contribute greatly to increasing the carrier concentration and mobility of in-based TFTs. However, its practical application is limited by the instability of In₂O₃ TFTs due to excessive oxygen vacancies ($V_O$). Therefore, it is essential for large-scale applications to perfect the stability of In₂O₃ TFTs. To date, several doping elements have been reported to improve the stability of devices, including Al [8], Ga [9], and Mg [10], which could suppress the formation of $V_O$. To obtain stable indium oxide TFTs, the doping element and oxygen should have a higher bond dissociation energy than In-O (360 kJ/mol). Among the many doping elements, titanium (Ti-O) has a larger oxygen bonding dissociation energy (662 kJ/mol) [6] than indium (In-O), which can reduce the $V_O$ concentration in the In₂O₃ film and be an effective oxygen vacancy inhibitor.

Among the methods for depositing oxide thin films, magnetron sputtering [11], electrospinning [12], sol-gel [13], etc. are not satisfactory in terms of uniformity of film formation and control of film thickness. Atomic layer deposition (ALD) technology offers

many advantages over the above preparation methods, including excellent repeatability, low deposition temperatures, and high control of composition and thickness [14–16]. As a result of the self-limiting nature of the ALD process, thin oxide films with precise thickness and conformability can be formed. For instance, Bak et al. [17] prepared IZTO TFTs with high-k $HfO_2$ as the dielectric layer by atomic layer deposition. Compared with $HfO_2$ dielectric devices prepared by magnetron sputtering, IZTO TFT samples prepared by the ALD method have a higher $I_{on}/I_{off}$ ratio and slightly improved field effect mobility. Yang et al. [18] fabricated 5 nm ultrathin $ZrO_2$ dielectrics by ALD for high-performance ZnO TFTs. The TFTs exhibit the lowest operating voltage of 1 V, field effect mobility of 36.8 $cm^2$/Vs, and *SS* of 69 mV/dec., which is close to the theoretical limit. Sheng et al. [19] successfully fabricated IGZO TFTs by PEALD with mobility up to 70 $cm^2$/Vs. As shown above, ALD processes have the potential to fabricate oxide TFTs with many advantages.

In this work, InTiO TFTs were fabricated using the ALD technique. Through the application of the proper amount of $Ti^{4+}$ doping, we have demonstrated improved electrical performance and stability of InTiO TFTs due to a reduction in $V_O$ defects and the density of interface traps.

## 2. Experiments

Figure 1a shows TFT fabricated on a 0.45-mm-thick p-type silicon wafer substrate with a bottom gate and top contact. Prior to preparation, the silicon substrates were washed sequentially with acetone, alcohol, and deionized water for 15 min in a sonicator, followed by 10 min of ozone and ultraviolet light treatment to enhance hydrophilicity. Subsequently, the 50-nm $Al_2O_3$ dielectric was deposited by the thermal ALD (TFS-200, Beneq) method at 250 °C with trimethylaluminum (TMA) and deionized water as aluminum and oxygen precursors, respectively. The optimized ALD cycle consists of a 200-ms precursor pulse combined with a 10-s $N_2$ purge. The InTiO films were grown on $Al_2O_3$ for the active layer at a growth temperature of 200 °C. High-purity (3-dimethylaminopropyl) dimethylindium (DADI) and tetrakis (dimethylamino) titanium (TDMATi) were used as precursors for indium and titanium, respectively, and ozone and deionized water were used as oxygen sources. The DADI and TDMATi canisters were heated to 35 and 45 °C, respectively. The pulse times of DADI and $O_3$ were 200/300 ms, and the pulse times of TDMATi and $H_2O$ were 500/250 ms. The nitrogen purging time is 5 s. *X* cycles of DADI/$O_3$ followed by one cycle of TDMATi/$H_2O$ were repeated for $Ti^{4+}$ incorporation into the $In_2O_3$ film, where *X* varied between 20, 18, 15, and 10, and films with approximately 17 nm thickness were obtained by adjusting the total supercycles. Thus, the grown InTiO thin films with different $In_2O_3$ sublayer cycles are named from InTiO (20:1) to InTiO (10:1). Figure 1b presents the detailed growth conditions, and Figure 1c shows the cross-sectional TEM image of the InTiO TFT. The pure $In_2O_3$ film was also prepared by ALD for contrastive analysis. After the active layer was deposited, 200-nm Al films were deposited by thermal evaporation as the source/drain electrodes. The channel width (*W*) and channel length (*L*) were 1000 and 100 μm, respectively. The devices were then annealed in the air for 10 min at 250 °C. The same process was used to grow InTiO thin films on quartz glass substrates to investigate their optical properties.

Through X-ray diffraction (XRD; Rigaku D/max-rB, Akishima, Japan) and X-ray photoelectron spectroscopy (XPS; Thermo Scientific K-Alpha+, Waltham, MA, USA), the crystal structure and chemical composition of the thin films were characterized. Atomic force microscopy (AFM, nanonaviSPA-400 SPM, SII Nano Technology Inc. Chiba City, Japan) was used to observe surface morphology, and a double-beam spectrophotometer (Hitachi U-3900H, Tokyo, Japan) was used to measure optical transmission. The semiconductor characteristics of TFT were measured by a semiconductor parameter analyzer (Keithley, 4200, Cleveland, OH, USA) and probe station (Lakeshore, TTP4, Carson, CA, USA). The electrical properties were tested by the Hall effect (Lakeshore 8400, Toyo Corporation, Tokyo, Japan).

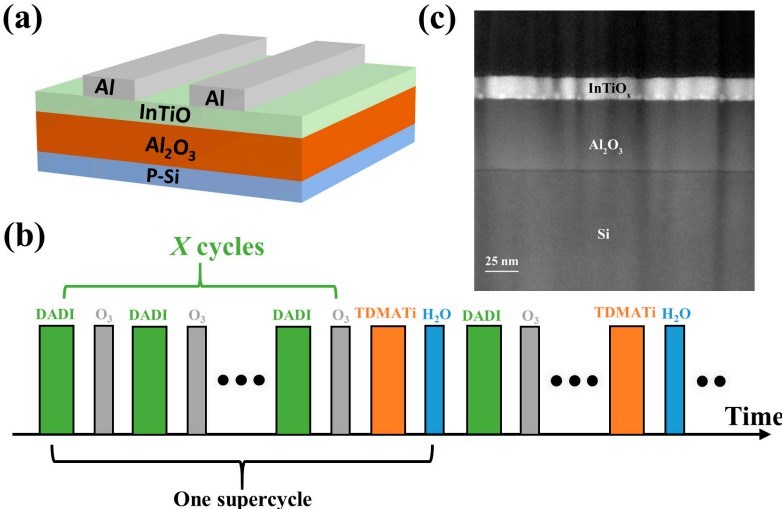

**Figure 1.** (**a**) The schematic structure of the InTiO TFT. (**b**) The cycle design of the ALD process for the thin films. (**c**) Cross-sectional TEM image of an InTiO TFT.

## 3. Results and Discussion

To evaluate the morphological evolution of the films with different composition ratios, the resulting InTiO films were characterized by AFM in the region of 2 μm × 2 μm, as shown in Figure 2a–d. AFM micrographs show a systematic decrease in surface roughness with decreasing number of *X* cycles (higher $Ti^{4+}$ doping ratio), which is well displayed by the obtained root mean squared (RMS) values due to the decrease in crystallinity of the film caused by $Ti^{4+}$ implantation. The values of RMS for InTiO (20:1), InTiO (18:1), InTiO (15:1), and InTiO (10:1) are 0.32, 0.28, 0.27, and 0.26 nm, respectively. The roughness of the active layer is one of the important parameters for ensuring high-performance thin-film transistor devices [20]. Its smooth surface is beneficial to source and drain electrode adhesion [21], thereby reducing the contact resistance between the electrode and the active layer. Contact resistance will be discussed later. All InTiO films have very flat surface morphology, which will help to reduce carrier scattering and promote mobility. Amorphous structures are observed in the XRD patterns of InTiO thin films, as shown in Figure 2e. TFT devices can be manufactured with high uniformity and smooth surfaces thanks to the amorphous phase [22]. The results are consistent with AFM analysis.

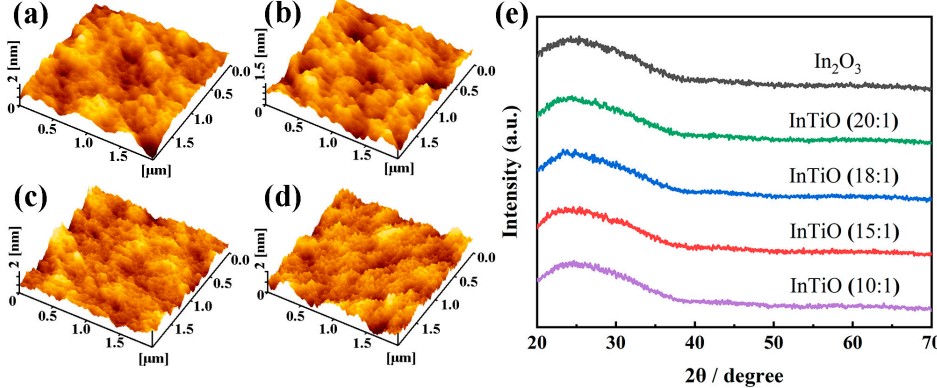

**Figure 2.** AFM images of (**a**) 20:1, (**b**) 18:1, (**c**) 15:1, (**d**) 10:1 InTiO thin films, and (**e**) XRD patterns.

The optical transmission spectra of all films on quartz glass substrates were measured in order to determine the effect of different $Ti^{4+}$ doping ratios on the optical band gap ($E_g$) of films. As shown in Figure 3a, all samples in the visible range have a transmittance

of about 90%, which makes them suitable for use in transparent electronic devices. The formula for $E_g$ and absorption coefficient ($\alpha$) is as follows:

$$\alpha h\nu = (h - E_g)^{\frac{1}{2}} \tag{1}$$

where $\alpha$, $h$, $\nu$, and $E_g$ are the optical absorption coefficient, the Planck constant, the incident photon frequency, and the optical energy bandgap, respectively. The absorption coefficient ($\alpha$) can be calculated using the following equation:

$$\alpha = (\frac{1}{d})\ln(\frac{1}{T}) \tag{2}$$

where $d$ is the film thickness and $T$ is the transmittance. Figure 3b shows the relationship between $(\alpha h\nu)^2$ and photon energy (eV). The $E_g$ is calculated to be 3.71, 3.72, 3.72, and 3.75 eV of InTiO (20:1), InTiO (18:1), InTiO (15:1), and InTiO (10:1), respectively. It is worth mentioning that the $E_g$ of pure $In_2O_3$ is 3.69 eV, which is mentioned in the introduction. As the $Ti^{4+}$ doping ratio increases, the optical band gap of thin film increases gradually. The narrower the band gap of a semiconductor, the easier valence band electrons transition to the conduction band to form electron-hole pairs, which causes higher carrier concentrations and results in devices that do not work properly [23]. With increasing $Ti^{4+}$ doping, the broadening of $E_g$.

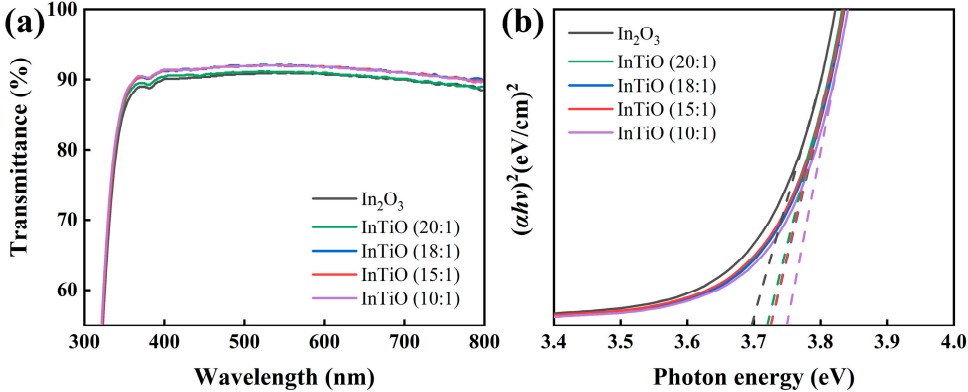

**Figure 3.** (**a**) The optical transmission spectra and (**b**) the optical bandgap of thin films deposited on quartz glass.

The chemical compositions of the InTiO thin films were investigated by the XPS. Figure 4 shows the O 1s peaks of the InTiO thin films with various doping ratios. All of these have been corrected by C 1s (284.5 eV). The O 1s of the XPS spectra were divided into three peaks centered at 529.8 ($O_I$), 531.3 ($O_{II}$, and 532.3 ($O_{III}$) eV, respectively. The $O_I$ sub-peak is attributed to the oxygen in oxide lattices (M-O), the $O_{II}$ sub-peak is associated with the oxygen bonds near $V_O$ in the lattice matrix of metal oxide, and the $O_{III}$ sub-peak can be assigned to hydroxyl species (M-OH) [24]. The $O_{II}$ / $O_I + O_{II} + O_{III}$ values of InTiO (20:1), InTiO (18:1), InTiO (15:1), and InTiO (10:1) are 39.09%, 37.26%, 36.15%, and 33.21%, respectively. It was found that the proportion of $V_O$ decreased with increasing $Ti^{4+}$ doping concentrations, which indicates that $Ti^{4+}$ doping can effectively reduce the amount of $V_O$, thus enhancing stability. Under ambient conditions after annealing at 250 °C for 10 min, the transportation characteristics of InTiO TFTs were measured to investigate the effect of $Ti^{4+}$ doping.

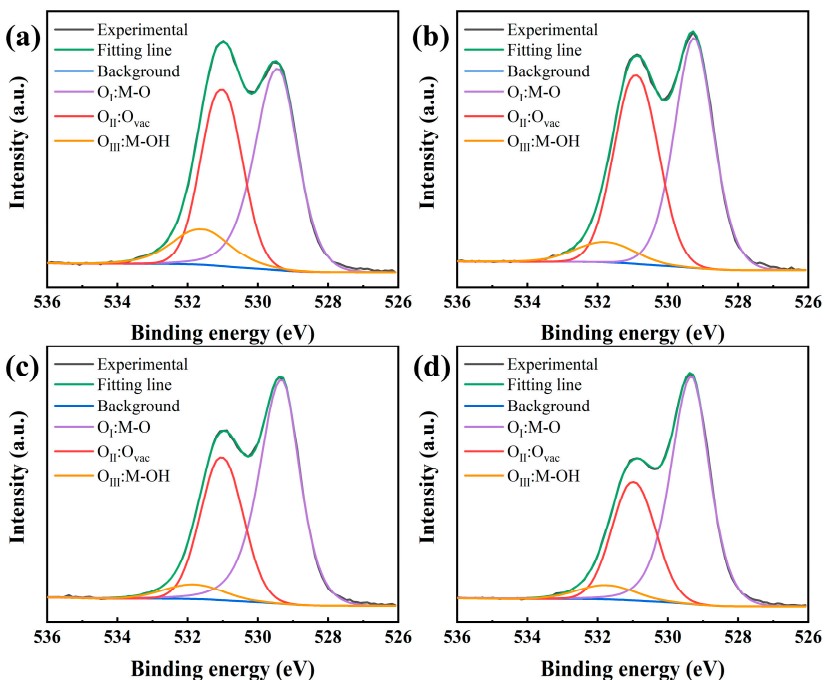

**Figure 4.** XPS O 1s of InTiO thin films with various composition ratios of (**a**) 20:1, (**b**) 18:1, (**c**) 15:1, and (**d**) 10:1.

As shown in Figure 5a, the source-drain voltage ($V_D$) is fixed at 5 V, while the gate-source voltage ($V_G$) increases from −5 to 15 V. As $V_G$ increases, the drain current ($I_D$) increases, indicating that InTiO semiconductors are n-type. Our previous work describes the method of determining $\mu$ and subthreshold swing ($SS$) [25].

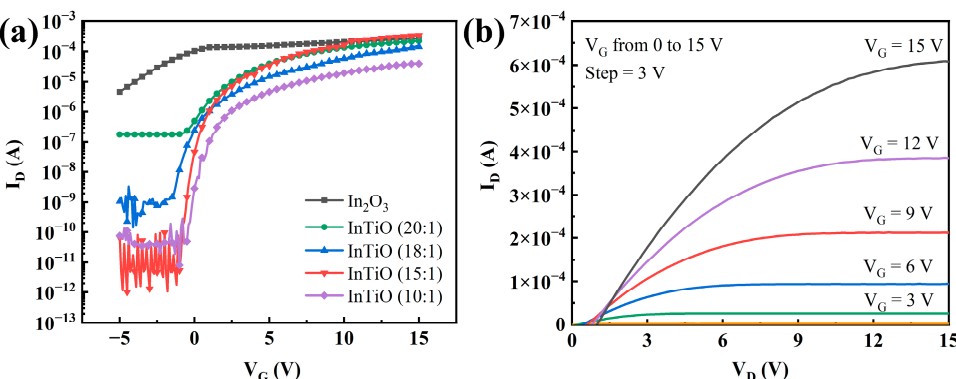

**Figure 5.** (**a**) Transfer characteristics ($V_D$ = 5 V) of TFTs with various $Ti^{4+}$ doping concentrations. (**b**) Output characteristics of TFTs based on InTiO (15:1).

The key electrical parameters, including $V_{th}$, mobility ($\mu$), $I_{on}/I_{off}$, and sub-threshold swing ($SS$), were presented in Table 1. Pure $In_2O_3$ TFTs did not exhibit semiconductor transfer characteristics and showed an on-conducting state. With the $Ti^{4+}$ doping ratio dropping from 20:1 to 10:1, $V_{th}$ increases from −1.12 to 0.29 V and $\mu$ decreases from 9.38 to 1.26 $cm^2/Vs$. The $\mu$ is affected by shallow traps near the conduction band and interfacial oxygen vacancies [26]. In addition, the value of $SS$ drops sharply from 1.43 to 0.13 V/dec. when the $Ti^{4+}$ doping rate continues to increase to 10:1, the $SS$ increases to 0.25 V/dec. The value of $SS$ is related to trapping states ($N_{trap}$); $N_{trap}$ is extracted using the following equation [27]:

$$N_{trap} = \left[ \frac{SS \log(e)}{kT/q} - 1 \right] \frac{C_i}{q} \tag{3}$$

where $k$ is Boltzmann's constant, $T$ is the temperature in Kelvin, and $q$ is the electron charge. It can be seen from the equation that the reason for the increase in $SS$ is attributed to the generation of more trap states due to excessive $Ti^{4+}$ doping.

**Table 1.** Electrical parameters of InTiO TFTs with different $Ti^{4+}$ doping ratios.

| Devices | $V_{th}$ (V) | $\mu$ (cm$^2$/Vs) | $I_{on}/I_{off}$ | $SS$ (V/dec.) |
|---------|--------------|-------------------|------------------|---------------|
| InTiO (20:1) | −1.12 | 9.38 | $10^3$ | 1.43 |
| InTiO (18:1) | −0.52 | 8.73 | $10^5$ | 0.74 |
| InTiO (15:1) | 0.17 | 7.69 | $10^7$ | 0.13 |
| InTiO (10:1) | 0.29 | 1.26 | $10^5$ | 0.25 |

The output characteristics of InTiO (15:1) TFTs are shown in Figure 5b. It can be seen from the figure that $I_D$ saturates at high source/drain bias ($V_D$). The pinch-off behavior and the saturation of the $I_D$ indicate that the contact between the source/drain electrodes and the active layer is excellent.

In order to explore the electrical properties of the films, we measured the electrical properties of InTiO$_x$ films on glass substrates. As presented in Figure 6a, all films exhibit a carrier concentration greater than $10^{18}$ cm$^{-3}$. The Hall mobilities of InTiO (20:1), InTiO (18:1), InTiO (15:1), and InTiO (10:1) are 9.2, 5.6, 3.9, and 1.2 cm$^2$/Vs, respectively. Notably, the Hall mobility of oxide semiconductors is partially proportional to carrier concentration [28]. Further, the resistivity of the film increases from 0.02 to 0.9 $\Omega \cdot$cm. We attribute this to $Ti^{4+}$ doping filling the oxygen vacancy defects, leading to reduced film conductivity.

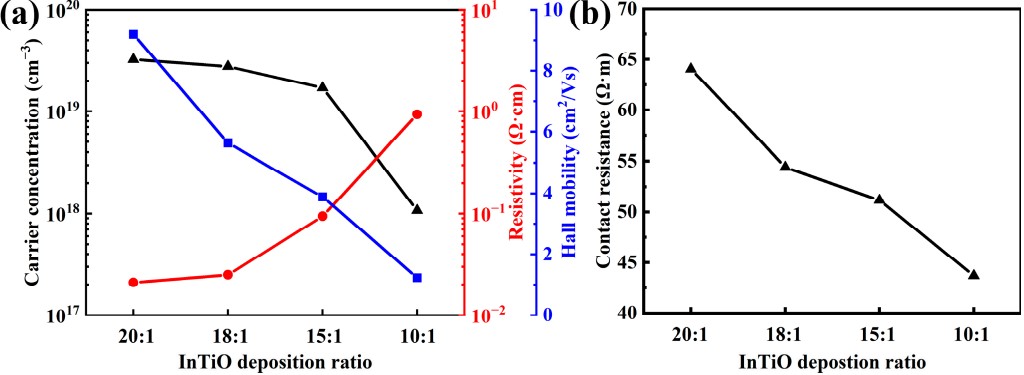

**Figure 6.** (a) The carrier concentration, resistivity, and Hall mobility of InTiO$_x$ films deposited on glass substrates. (b) The changing trend of contact resistance with doping ratios.

The contact resistance between the source-drain electrode and the active layer can be extracted by the transmission line method (TLM) [29,30]. In this method, the total resistance is obtained by adding the resistance of the active layer and the contact resistance between the source and drain electrodes, as shown in the following equation:

$$R_T = \frac{V_D}{I_D} = R_{SD} + \frac{L}{\mu C_{ox} W (V_G - V_{th})} \tag{4}$$

$R_T$ represents the total resistance of the InTiO$_x$ TFT, while $R_{SD}$ represents the contact resistance between the source and drain electrodes and the active layer. Using the equation above, we calculate the contact resistances of InTiO (20:1), InTiO (18:1), InTiO (15:1), and InTiO (10:1) at a gate voltage of 15 V, which are 64, 54, 51, and 45 $\Omega \cdot$m, respectively. The relationship between contact resistance and doping ratio is shown in Figure 6b. With the increase in doping ratio, contact resistance gradually decreases. InTiO (10:1) has the smallest contact resistance, which is consistent with the analysis of AFM results.

As shown in Figure 7a–c, the latter three devices were selected for forward bias stability testing. At room temperature and dark conditions, the transfer curves of 900 s

were measured under +5 V positive bias stress (PBS). The results show that the transfer characteristic curves of the three devices have a certain degree of positive shift. The stability of thin film transistors is related to the interface state of the device's dielectric layer and active layer and the $V_O$ inside the active layer [31]. The previous XPS results show that $Ti^{4+}$ doping leads to a decrease in oxygen-related vacancies in the film, thereby improving the stability of the TFT device. The results were consistent with the XPS analysis.

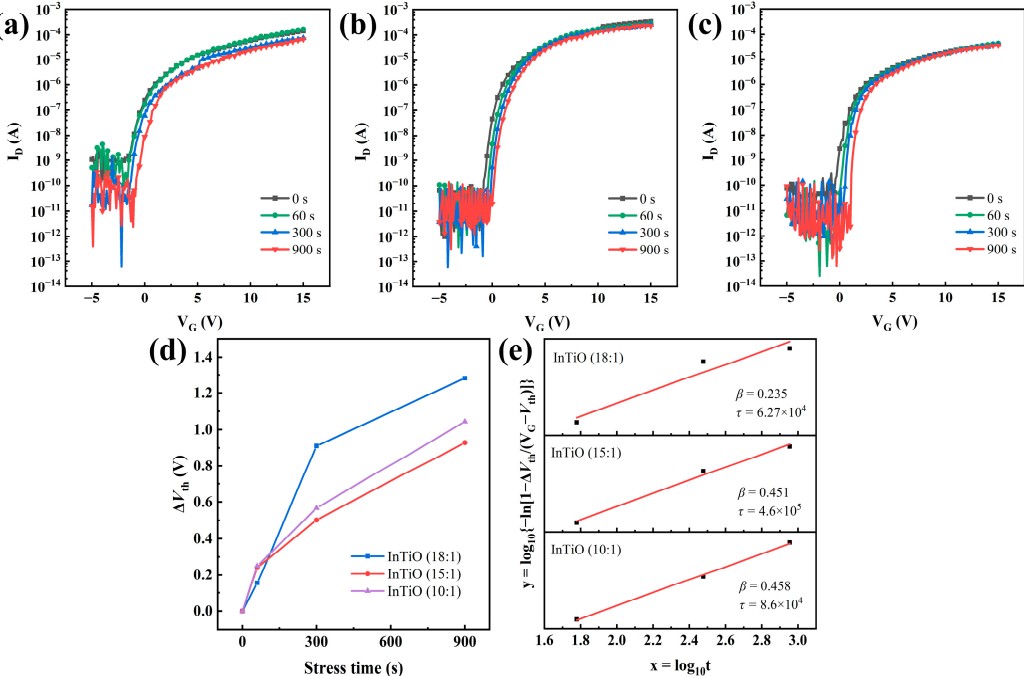

**Figure 7.** PBS of InTiO TFTs with various $Ti^{4+}$ concentrations, (**a**) 18:1, (**b**) 15:1, and (**c**) 10:1. (**d**) The dependence of $\Delta V_{th}$ on the stress time of InTiO TFT devices. (**e**) The fitted curves described by Equation (5).

Under various PBS time intervals, Figure 7d displays the threshold voltage shift of InTiO TFTs. As mentioned above, the $\Delta V_{th}$ of InTiO (15:1) is only 0.93 V. Compared to this device, the $\Delta V_{th}$ of InTiO (18:1) and InTiO (10:1) are 1.29 and 1.04 V. According to the result, $Ti^{4+}$ doping can improve transistor stability with the appropriate amount. $\Delta V_{th}$ can be calculated by the following equation:

$$\Delta V_{th} = (V_G - V_{th})\left\{1 - \exp\left[-\left(\frac{t}{\tau}\right)^{\beta}\right]\right\} \tag{5}$$

where $\tau$ is the characteristic of carrier trapping time and $\beta$ is the stretching exponent. The $V_{th}$ shift is proportional to the value of $V_G$-$V_{th}$, implying that $\Delta V_{th}$ is related to $V_{th}$, but $\tau$ and $\beta$ hardly depend on the bias stress amplitude. The values of $\tau$ and $\beta$ extracted from the equation are $6.27 \times 10^4$, $4.6 \times 10^5$, and $8.6 \times 10^4$, and 0.235, 0.451, and 0.458 for InTiO TFTs with $Ti^{4+}$ doping of 18:1, 15:1, and 10:1, respectively. The fitted curve described by Equation (5) is shown in Figure 7e. This demonstrates that under extended positive bias testing, InTiO (15:1) TFTs can resist greater performance drift, and it can be obtained that InTiO TFTs can be effectively with appropriate amounts of $Ti^{4+}$ doping.

## 4. Conclusions

In this paper, we report the successful preparation of InTiO TFTs using the ALD method. The microstructure and defect characteristics of InTiO films were investigated through XRD and XPS analyses. All films exhibit a low RMS value of less than 1 nm and are amorphous. Additionally, all InTiO films display an average transmittance of over 90%

in the visible range. Notably, the InTiO (15:1) TFTs exhibit exceptional stability due to a reduction in oxygen vacancy defects to 33.21%, resulting in a minimum $\Delta V_{th}$ of 0.93 V. Our results indicate that an appropriate amount of $Ti^{4+}$ doping effectively inhibits oxygen defects in the active layer. Consequently, these InTiO TFTs with superior electrical properties and high transparency are promising candidates for use in the field of transparent displays.

**Author Contributions:** B.Y., methodology, investigation, validation, and writing—original draft; P.L., methodology, investigation, and writing—review & editing; Z.C., investigation, formal analysis, and visualization; H.X., investigation, formal analysis, and visualization; C.F., conceptualization, project administration, and data curation; X.D., conceptualization, project administration, supervision, and writing—review & editing; J.Z., data curation, supervision, and funding acquisition. All authors have read and agreed to the published version of the manuscript.

**Funding:** This work is supported by the National Natural Science Foundation of China (62274105), and C.F. acknowledges the support of the National Natural Science Foundation of China (21902063).

**Institutional Review Board Statement:** Not applicable.

**Informed Consent Statement:** Not applicable.

**Data Availability Statement:** Not applicable.

**Conflicts of Interest:** The authors declare no conflict of interest.

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
