# Peer review of "Effect of Titanium Cation Doping on the Performance of In2O3 Thin Film Transistors Grown via Atomic Layer Deposition"

_coatings, doi:10.3390/coatings13030605_

Round 1

Reviewer 1 Report

In this manuscript, Yang et. al incorporating titanium cation (Ti4+) into In2O3 via atomic layer deposition to reduce the carrier concentration and excess oxygen defects in In2O3. Thin film transistors (TFTs) with In:Ti atomic ratio of 15:1 demonstrated lower threshold voltage, decreased subthreshold swing, higher Ion/Ioff ratio and larger transmittance compared with pristine In2O3. The excellent performance exhibite great potential for real applications in microelectronic devices. The manuscript is well documented and well written. There are only a few remarks that need to be addressed by the authors prior to publication:

1. Please pay attention to the format issue in the main text. For example, the unit of y axis in Figure 3(b).

2. “All InTiO films have very flat surface morphology, which will help to reduce carrier scattering and promote mobility”. Please provide the electrical properties (such as concentration and hall mobility) to verify it.

3. Authors claim “Its smooth surface is beneficial to source and drain electrodes adhesion”. Can you provide a contact resistance trend with Ti doping ratio?

4. Figure 5. (b), can you explain why diode behavior is getting more appeared as VG increases?

Reviewer 2 Report

This report presents the “Effect of titanium cation doping on the performance of In2O3 thin film transistors grown via atomic layer deposition”. In this work, the authors have used the ALD method to dope InOx with Ti and found the optimized process condition. The paper is worth publishing although the text is in some places stiff and harping.

- The authors compared the various properties of TFT devices, but there have been many reports about the InOx TFT devices. What is the originality of this manuscript?

--Please enrich the introduction with the help of: (https://doi.org/10.1016/j.tsf.2022.139290), (https://doi.org/10.1016/j.apsusc.2021.150152).

-“Among the methods for depositing oxide thin films, magnetron sputtering [11], electrospinning [12], and sol-gel [13], etc. are not satisfactory in terms of uniformity of film formation and control of film thickness”. thin films from electrospinning is possible?

-The authors need to revise the abbreviation of dimethylindium. It could be DMIn not DADI.

-Why authors have used two different reactants?

- There is a lack of visible evidence such as SEM and TEM analysis. To support the schematic structure of the InTiO TFT, the surfaces and cross-sectional view of the films should be assessed.

- The author may suggest revising the conclusion into brief. The abstract and the conclusion should follow their subject with data, numbers, etc., not just motivations and summary.

Round 2

Reviewer 2 Report

The authors have improved the manuscript significantly and thus can be considered. Thank you